# Modulation of TRPV1 and TRPA1 Channels Function by Sea Anemones’ Peptides Enhances the Viability of SH-SY5Y Cell Model of Parkinson’s Disease

**DOI:** 10.3390/ijms25010368

**Published:** 2023-12-27

**Authors:** Yuliya S. Kolesova, Yulia Y. Stroylova, Ekaterina E. Maleeva, Anastasia M. Moysenovich, Denis V. Pozdyshev, Vladimir I. Muronetz, Yaroslav A. Andreev

**Affiliations:** 1Shemyakin-Ovchinnikov Institute of Bioorganic Chemistry, Russian Academy of Sciences, 117997 Moscow, Russia; vasilenko-yuliya@mail.ru (Y.S.K.); katerina@1ns.ru (E.E.M.); a-moisenovich@mail.ru (A.M.M.); 2Institute of Molecular Medicine, Sechenov First Moscow State Medical University, 119991 Moscow, Russia; stroylova@belozersky.msu.ru; 3Belozersky Institute of Physico-Chemical Biology, Lomonosov Moscow State University, 119991 Moscow, Russiavimuronets@belozersky.msu.ru (V.I.M.); 4Department of Biology, Lomonosov Moscow State University, 119991 Moscow, Russia

**Keywords:** ion channel, TRPA1, TRPV1, ASIC1A, alpha-synuclein, cell model, neuroblastoma SH-SY5Y, neuronal differentiation, Parkinson’s disease

## Abstract

Cellular dysfunction during Parkinson’s disease leads to neuroinflammation in various brain regions, inducing neuronal death and contributing to the progression of the disease. Different ion channels may influence the process of neurodegeneration. The peptides Ms 9a-1 and APHC3 can modulate the function of TRPA1 and TRPV1 channels, and we evaluated their cytoprotective effects in differentiated to dopaminergic neuron-like SH-SY5Y cells. We used the stable neuroblastoma cell lines SH-SY5Y, producing wild-type alpha-synuclein and its mutant A53T, which are prone to accumulation of thioflavin-S-positive aggregates. We analyzed the viability of cells, as well as the mRNA expression levels of TRPA1, TRPV1, ASIC1a channels, alpha-synuclein, and tyrosine hydroxylase after differentiation of these cell lines using RT-PCR. Overexpression of alpha-synuclein showed a neuroprotective effect and was accompanied by a reduction of tyrosine hydroxylase expression. A mutant alpha-synuclein A53T significantly increased the expression of the pro-apoptotic protein BAX and made cells more susceptible to apoptosis. Generally, overexpression of alpha-synuclein could be a model for the early stages of PD, while expression of mutant alpha-synuclein A53T mimics a genetic variant of PD. The peptides Ms 9a-1 and APHC3 significantly reduced the susceptibility to apoptosis of all cell lines but differentially influenced the expression of the genes of interest. Therefore, these modulators of TRPA1 and TRPV1 have the potential for the development of new therapeutic agents for neurodegenerative disease treatment.

## 1. Introduction

The accumulation of protein aggregates in Lewy bodies in dopaminergic neurons in Parkinson’s disease is accompanied, also, by disturbances in energy metabolism, impairment of mitochondrial function, oxidative stress, and a dysfunctional mechanism of protein degradation [1]. Such cellular dysfunction leads to neuroinflammation in various brain regions, further contributing to the progression of the disease [2,3]. Patients suffer from a range of secondary motor and non-motor symptoms in addition to the classic symptoms of affection of the nigrostriatal dopaminergic pathways [4]. The presence of abnormal forms of α-synuclein (α-syn) in the form of neurotoxic oligomers and fibrils is one of the key pathogenic features of Parkinson’s disease (PD). A missense mutation in the alpha-synuclein gene corresponding to the A53T substitution was found in a family study of patients with autosomal dominant PD [5]. As there are currently no neuroprotective therapies that can delay or prevent the progression of Parkinson’s disease (PD), elucidating new potential therapeutic targets and treatment approaches is a major healthcare and societal challenge [6].

Transient receptor potential vanilloid-1 (TRPV1) and ankyrin-1 (TRPA1) ion channels and acid-sensing ion channel 1a (ASIC1a) are highly expressed throughout the brain, where they are involved in the regulation of many important physiological and pathological processes [7,8,9]. Numerous studies have implicated these channels in the pathogenesis of Parkinson’s disease, but understanding the mechanisms is still far from complete.

Recent studies strongly suggest that dysregulation of the TRP channel functions is involved in various pathological events in neurodegenerative disorders [7]. Several studies have indicated that TRPV1 is involved in pain mechanisms in Parkinson’s disease. This pain can be nociceptive or neuropathic in nature, and it has been shown in a 6-OHDA-lesioned mice model that blocking TRPV1 resulted in pain relief in animals [10]. 

There is growing evidence that acid-sensing ion channels (ASICs) play a functional role in neuronal differentiation, where they modulate membrane excitability, maturation of dendrites and neurites, Ca^2+^ homeostasis, and dopamine secretion in dopaminergic neurons [8,11,12]. 

APHC3 has been reported as an analgesic peptide acting on the TRPV1 channel, with different modes of action depending on the type and strength of the stimuli. In whole-cell patch clamp experiments, it significantly inhibited the response of TRPV1 to high concentrations of agonists (IC_50_ 18 nM) and potentiated the response to low concentrations of agonists [13]. Administration of this peptide in mice leads to the analgesic and anti-inflammatory effect, which corresponds to inhibition or desensitization of TRPV1-positive sensory neurons [14,15,16], and a decrease in core body temperature, which is a hallmark of TRPV1 agonists or positive modulators of response to low pH stimuli [14].

Ms 9a-1 was found to be a positive modulator of the TRPA1 channel (EC_50_ 32–210 nM), with significant analgesic and anti-inflammatory activity [17,18,19]. Activation of TRPA1 on sensory neurons is the crucial point of the analgesic action of Ms 9a-1, and pretreatment of experimental animals with a selective TRPA1 antagonist totally reverses the analgesic effect of the peptide [17]. Several weak activators of TRPA1 [20,21,22] can produce a similar effect by desensitizing the channel and decreasing the ability of TRPA1-expressing neurons to respond to other stimuli.

In the present work, we investigated the neuroprotective properties of two peptides, Ms 9a-1 and APHC3, in differentiated SH-SY5Y cells, a model that is more appropriate and widely used for experimental research on neurodegenerative diseases such as Parkinson’s disease. A two-step differentiation into dopaminergic phenotype protocol with retinoic acid (RA) and further addition of brain-derived neurotrophic factor (BDNF) [23,24,25] was applied to stable neuroblastoma cell lines SH-SY5Y, which produce wild-type alpha-synuclein, and its mutant A53T, which is prone to accumulate thioflavin-S-positive aggregates [26]. We analyzed the viability of the cells and the rate of late apoptosis/necrosis (cell death) and the change in mRNA expression in this model of several genes of interest upon cell differentiation and peptide treatment, which are primarily the genes for alpha-synuclein (SNCA), tyrosine hydroxylase (TH), BAX, and bcl2, as well as ASIC1a, TRPV1 and TRPA1.

## 2. Results

### 2.1. Analysis of Neuronal Differentiation with RA + BDNF of Overexpressing Alpha-Synuclein SH-SY5Y Cells 

We used three cell lines for differentiation into dopaminergic neuron-like cells using the RA + BDNF protocol: native neuroblastoma SH-SY5Y; cells SH-SY5Y stably producing wild-type alpha-synuclein (α-synWT); and cells SH-SY5Y stably producing mutant A53T alpha-synuclein (α-synA53T). To confirm that differentiated cells express mature neuronal markers, we performed immunocytochemistry for specific markers: β-III tubulin and NCAM (neural cell adhesion molecule, CD56), which are ubiquitously expressed in the nervous system. After RA + BDNF differentiation, expression of the neuronal markers NCAM and β-III tubulin was observed in all the investigated cell lines (Figure 1). During differentiation, we observed the growth of neurites in the cells and the formation of a dense neuron-like network over a period of 10 days.

In addition, all differentiated cell lines showed tyrosine hydroxylase (TH) expression, which is often used as a marker for dopaminergic neuron-like cells. TH expression was not detected in undifferentiated cells, whereas all differentiated cells showed TH expression, which confirmed successful differentiation in dopaminergic-like phenotypes (Figure 2). According to the reverse transcription-quantitative polymerase chain reaction (RT-qPCR) results, the relative levels of TH are more strongly expressed in differentiated SHSY5Y cells than in differentiated α-synWT and α-synA53T. 

### 2.2. Quantitative mRNA Difference in Alpha-Synuclein (SNCA) and Tyrosine Hydroxylase (TH) in Overexpressing Alpha-Synuclein SH-SY5Y after Differentiation with RA + BDNF

To describe the changes in SH-SY5Y cells after differentiation with RA + BDNF, we determined the levels of mRNA transcripts for the genes of alpha-synuclein (SNCA) and tyrosine hydroxylase (TH) on day 10. Reverse transcription followed by qPCR experiments showed that the level of SNCA expression in α-synWT and α-synA53T cells is about 1.5–1.7 times higher compared with the control SH-SY5Y cells (Figure 3, Table 1). Interestingly, analysis of TH expression revealed that compared to differentiated SH-SY5Y neuroblastoma, overexpression of alpha-synuclein resulted in a 72% decrease in TH mRNA levels for the α-synWT and a 94% decrease in the α-synA53T cells (Figure 3).

### 2.3. Quantitative mRNA Difference in TRPA1, TRPV1, and ASIC1a in Overexpressing Alpha-Synuclein SH-SY5Y Cells after Differentiation with RA + BDNF

As reported earlier, SH-SY5Y cells express functional ASIC1a, TRPV1, and TRPA1 channels [27]. To confirm the suitability of the differentiated neuroblastoma cell model for testing peptides that modulate TRPV1 and TRPA1 channel activity, we determined the levels of mRNA transcripts for TRPV1, TRPA1, and ASIC1a after 10 days of differentiation with RA + BDNF. Reverse transcription followed by qPCR experiments showed the presence of all transcripts of interest. The number of TRPA1 mRNA transcripts was approximately doubled in α-synWT and α-synA53T differentiated cells compared to differentiated SH-SY5Y cells (Figure 4). In the case of TRPV1, the expression level increased by 45% only in α-synA53T cells and did not change in the wild-type α-synWT cells (Figure 4). Finally, in both α-synWT and α-synA53T cells, ASIC1a showed no statistically significant change in expression compared to control cells (Figure 4).

### 2.4. Viability, Cell Death, and Apoptosis of Overexpressing Alpha-Synuclein SH-SY5Y Cells after Differentiation with RA + BDNF

We estimated viability as the number of metabolically active cells at day 1 (undifferentiated SH-SY5Y) and day 10 (differentiated, dopaminergic neuron-like cells) using the CCK-8 assay (Figure 5A, Table 1). Compared to SH-SY5Y cells, the level of cell viability was reduced by 25% in α-synWT cells (Figure 5A). However, in dopaminergic neuron-like cells, viability equalized in all groups and was even higher for α-synA53T cells by 13% compared to SH-SY5Y cells (Figure 5B).

We have also compared cell death in all differentiated cell lines. Cell death was determined before RA + BDNF treatment on day 1 (undifferentiated cells) and after treatment on day 10 (differentiated cells) by fluorescent propidium iodide staining to identify cells in late apoptosis and necrosis. We assessed the number of propidium iodide-stained cells twice, before and after freeze-thawing, to estimate the percentage of cells in late apoptosis and necrosis. Cell death is represented as the percentage of PI-positive cells among all cells detected after freezing and thawing. Following sequential RA + BDNF treatment, cells of all three lines exhibited increased cell death. Compared to undifferentiated cells, cell death increased ∼10-fold in SH-SY5Y cells, ∼3-fold in cells α-synWT, and ∼6-fold in α-synA53T cells (Figure 5C), which is in agreement with previous reports [28,29,30]. 

We analyzed the BAX/Bcl2 ratio using RT-qPCR to evaluate the susceptibility of different cell lines to apoptosis. Bcl-2 family proteins are among the major anti-apoptotic regulators in cells, while BAX family proteins oppose them, being pro-apoptotic. A high BAX/Bcl-2 ratio characterizes apoptosis susceptible cells [31,32]. Differentiated cells of all studied cell lines showed increased susceptibility to apoptosis compared to the corresponding undifferentiated cells (Figure 5D). However, differentiated α-synWT cells showed less susceptibility to apoptosis than differentiated SH-SY5Y cells and α-synA53T cells. In differentiated SH-SY5Y cells, susceptibility to apoptosis, as assessed by the BAX/Bcl2 ratio, was increased 5.7-fold, whereas it was increased only 1.8-fold in α-synWT cells (Figure 5D). During differentiation, the relative expression of the pro-apoptotic gene BAX was slightly downregulated in SH-SY5Y cells, but the expression of anti-apoptotic Bcl-2 was markedly reduced by 86% compared to undifferentiated cells (Figure 5E,F). Expression of anti-apoptotic Bcl-2 decreased in α-synWT cells to a lower degree, i.e., by 55%. However, the expression of proapoptotic BAX did not change compared to differentiated SH-SY5Y cells, which leads to an increased BAX/Bcl-2 ratio in differentiated α-synWT cells compared to undifferentiated. α-SynA53T cells demonstrated a 7-fold increased susceptibility to apoptosis during differentiation, due to an increase in the expression of proapoptotic BAX (5.5-fold) without the change in the expression of Bcl-2 (Figure 5E,F). 

### 2.5. Ms 9a-1 and APHC3 Peptides Ameliorate Cell Viability and Apoptosis Resistance of Neuron-like Cells 

We examined the effects of Ms 9a-1 and APHC3 peptide exposure on the cell model viability by their metabolic activity using the CCK-8 assay. The viability of peptide-treated cells was expressed as a percentage relative to the viability of untreated cells. 

Pretreatment of differentiated SH-SY5Y and α-synWT cells with 300 nM Ms 9a-1 for 24 h significantly increased cells viability to 16.37 ± 3.02% and 11.97 ± 2.06%, respectively (*p* < 0.005) (Figure 6A). In differentiated α-synA53T cells, the change in viability was not significant. 

When cells were pretreated with 300 nM APHC3 peptide for 24 h, a significant increase in cell viability was observed up to 46.05 ± 1.73% in differentiated SH-SY5Y cells and up to 36.12 ± 4.49% in differentiated α-synWT cells (*p* < 0.0001) (Figure 6A). In differentiated α-synA53T cells, pre-treatment with 300 nM APHC3 did not significantly increase viability compared to untreated α-synA53T cells (Figure 6A). 

We analyzed the effect of Ms 9a-1 and APHC3 treatment on dead cells’ populations in differentiated cell lines. In differentiated SH-SY5Y and α-synWT cells, pretreatment for 24 h with 300 nM Ms 9a-1 showed a statistically insignificant reduction in PI-positive cells. However, in the differentiated α-synA53T cells, Ms 9a-1 reduced the dead cell population by 14.35 ± 4.05% (Figure 6B). Treatment with APHC3 peptide resulted in a reduction in cell death of 13.97 ± 7.4% in differentiated SH-SY5Y cells and 14.06 ± 1.28% in α-synA53T cells and did not alter the dead cell population in the differentiated α-synWT cells (Figure 6B).

Peptides Ms 9a-1 and APHC3 significantly reduced the expression level of pro-apoptotic BAX in differentiated SH-SY5Y cells, while the expression level of anti-apoptotic Bcl-2 was almost unchanged, resulting in a significant decrease in the BAX/Bcl-2 ratio (by 35.4% with Ms 9a-1 and 72% with APHC3) (Figure 6C–E). Ms 9a-1 and APHC3 upregulated the expression level of BAX and Bcl-2 in differentiated α-synWT and α-synA53T cells (Figure 6D,E). Consequently, the BAX/Bcl-2 ratios were reduced by 21% (α-synWT) and 26.6% (α-synA53T) for Ms 9a-1 and by 24.5% and 13.7%, respectively, for APHC3 (Figure 6C). 

### 2.6. Quantitative mRNA Difference in TRPA1, TRPV1, and ASIC1a in Differentiated Overexpressing Alpha-Synuclein SH-SY5Y after Ms 9a-1 or APHC3 Treatment

The relative mRNA levels of ASIC1a, TRPA1, and TRPV1 were assessed to determine whether peptide treatment could affect their expression (Figure 7, Table 2). 

We did not observe statistically significant changes in ASIC1a expression levels after 24 h treatment by Ms 9a-1 (300 nM) and APHC3 (300 nM) (Figure 7), except for α-synWT cells, which showed a minimal (~16%) decrease in ASIC1a expression after Ms 9a-1 treatment (Figure 7B). The APHC3 peptide did not produce a significant effect on the expression level of ASIC1a in any of the cell lines 24 h after treatment.

Ms 9a-1 produced a significant decrease in TRPA1 expression: by 79% in differentiated SH-SY5Y and by 44.7% in α-synA53T cells. However, it had no statistically significant effect in α-synWT cells. Simultaneously, APHC3 treatment significantly reduced TRPA1 expression levels in differentiated SH-SY5Y (82.5%), α-synWT (24%), and α-synA53T (58%) cells.

Ms 9a-1 downregulated TRPV1 expression by 57% compared to untreated cells in differentiated SH-SY5Y (Figure 7A) but had no effect on differentiated α-synWT and α-synA53T cells (Figure 7B,C). After 24 h treatment with 300 nM APHC3, expression of TRPV1 was downregulated by 77% in differentiated SH-SY5Y, upregulated (~54%) in α-synWT cells, and no effect was observed in α-synA53T cells.

All the changes in differentiated cells induced by Ms 9a-1 and APHC3 treatment are summarized in Table 2.

## 3. Discussion

The importance of studying the mechanisms of cellular response to an increase in alpha-synuclein, especially its A53T mutant associated with Parkinson’s disease, is still very relevant. Data obtained with peptides acting on the TRPV1 and TRPA1 channels indicate their involvement in the response of neurons to stress caused by the accumulation of thioflavin-S-positive aggregates, leading to their death in neurodegeneration. The accumulation of misfolded forms and aggregates of alpha-synuclein and other proteins disrupts the function of several cellular systems and leads to a range of disorders including proteasome and autophagy dysfunction, mitochondrial dysfunction, and oxidative stress. 

According to current knowledge, membrane receptors and ion channels largely determine the functioning of a living cell, play a crucial role in the transmission of intercellular signals, and are involved in the development of various diseases and pathologies, including neurodegenerative diseases [33]. Regulation of the function of these receptors by specific ligands can therefore alter intracellular metabolism, thereby stimulating or slowing down pathological cell transformation. Bulk transcriptomic approaches, including single-cell data, are also appropriate to identify differences in gene expression patterns and the functional biological processes to which they are linked in PD pathogenesis [34]. One possibility for how the expression of genes involved in the pathogenesis of PD may be altered is at the level of transcriptional regulation, as has been shown for cerebral cavernous malformation disease, which also exists in sporadic and genetic forms [35].

In this work, we characterized a cell model of Parkinson’s disease based on SH-SY5Y neuroblastoma cells with stable alpha-synuclein overexpression, which were differentiated towards dopaminergic neuronal phenotype by sequential addition of RA + BDNF. We observed that during differentiation, which itself stresses neuroblastoma cells and makes them more susceptible to apoptosis, wild-type alpha-synuclein overexpression decreased cell death, which is consistent with one of the normal functions of alpha-synuclein being neuroprotective [36]. Mutant alpha-synuclein A53T lost the neuroprotective properties, significantly increasing the expression of the proapoptotic protein BAX (Figure 5). According to our previous data [26], α-synA53T cells accumulate significantly more thioflavin-S-positive aggregates than cells overexpressing wild-type synuclein. At the same time, a marked but relatively small increase in synuclein levels (expression increased 1.5-fold) was accompanied by a decrease in tyrosine hydroxylase levels. Both mRNA and protein levels of TH in dopamine neurons in the substantia nigra are known to be reduced in PD patients [37]. Therefore, overexpression of alpha-synuclein is a native neuroprotective mechanism that can lead to a decrease in dopamine production and boosting of PD symptoms. In addition, we found an increase in TRPA1 expression levels both in α-synWT and α-synA53T cells, and in the case of TRPV1 only in the α-synA53T cells, allowing us to adequately test the effect of their ligands on cell survival in this model. 

It is well known that the development of Parkinson’s disease is accompanied by a series of cell dysfunctions, culminating in the loss of neurons and brain function [38,39]; however, effective prevention and treatment strategies have not yet been identified. Therefore, we investigated therapeutic approaches to pathophysiological phenomena mediated by transient receptor potential ion channels, such as TRPA1 and TRPV1. 

TRPV1 is considered to play a major role in the disruption of calcium homeostasis under inflammatory conditions, which is a strong likelihood in the development of many neurodegenerative processes and in particular the pathogenesis of Parkinson’s disease [40,41,42,43]. The role of TRPV1 channels in PD is controversial [44]. Activation of TRPV1 mediates cell death of DA neurons. TRPV1 may contribute to neurodegeneration in response to endogenous ligands such as AEA [45]. However, when inhibition and activation of TRPV1 were studied in preclinical models of PD, both approaches possessed beneficial outcomes [44].

TRPA1 is a neuronal sensor of reactive oxygen species [46], while oxidative stress is considered to be one of the most important contributors to the death of substantia nigra cells in PD [47]. However, the role of TRPA1 in PD pathogenesis is under-investigated. Nevertheless, expression of TRPA1 in substantia nigra was reported [48,49]. Moreover, an important role of acrolein, an endogenous TRPA1 agonist, was found in PD pathology [49,50,51]. It is noteworthy that carvacrol (an agonist of TRPA1 from plants) might protect dopaminergic neurons in an animal model of PD [48].

APHC3 is a complex modulator of the TRPV1 channel. The action mode of APHC3 on the TRPV1 channel is bimodal and depends on the activation stimuli strength, it acts as a positive modulator of low-amplitude responses and inhibits high-amplitude responses [13]. APHC3 at 300 nM acted differently on three tested cell lines. It significantly decreased apoptosis of SH-SY5Y differentiated in dopaminergic neuron-like cells and α-synA53T neuron-like cells but was unable to improve the antiapoptotic effect of α-synWT overexpression (Figure 6). Nevertheless, APHC3 significantly reduced the BAX/Bcl-2 ratio, decreasing the cell susceptibility to apoptosis in all cell lines (Figure 6C). Intriguingly, APHC3 produces different effects on the expression of Bax, Bcl-2, TRPV1, and TRPA1 in differentiated SH-SY5Y and α-synWT/ α-synA53T cells (Figure 6 and Figure 7). Evidently, APHC3 increased cell viability in differentiated SH-SY5Y by suppression of BAX, TRPV1, and TRPA1 expression. In α-synWT/ α-synA53T cells, expression of both BAX and Bcl-2 was upregulated resulting in a decreased BAX/Bcl-2 ratio. 

Peptide Ms 9a-1 is the positive modulator of the TRPA1 channel, which is isolated from the sea anemone *Metridium senile*. Ms 9a-1 significantly potentiates agonist-induced currents of TRPA1 in vitro [17], but intravenous or subcutaneous injection of Ms 9a-1 (0.1–0.3 mg/kg) reduces pain, inflammation, and hyperalgesia in different models of pain [17,18], including MIA-induced osteoarthritis [19]. Ms9a-1 showed similar effects to APHC3 on all tested cell lines, which were mostly less pronounced. Only treatment with Ms 9a-1 of α-synA53T cells resulted in a more apparent decrease in the Bax/Bcl-2 ratio and consequently reduced cells’ susceptibility to apoptosis (Figure 7C, Table 2). This fact confirms the more important role of TRPV1 than TRPA1 in the homeostasis of dopaminergic neurons. Nevertheless, modulation of TRPA1 can significantly affect cell viability and apoptosis.

This study provides evidence that Ms 9a-1 and APHC3 may become novel candidate molecules for the treatment of neurodegenerative conditions, including mutant alpha-synuclein-induced neuronal injury. It is important to note the significant difference between all three tested cell lines in terms of expression of pro- and anti-apoptotic genes and response to modulators of TRPV1 and TRPA1 channels. Therefore, at least at the cellular level, mutant α-synA53T-induced PD differed from the PD of other etiology.

Despite confirming that the Ms 9a-1 and APHC3 peptides can protect dopaminergic neurons from α-synA53T -induced injury and apoptosis, this study has certain limitations. The function of the Ms 9a-1 and APHC3 peptides on alpha-synuclein-induced neurotoxicity in vivo needs to be further elucidated. 

## 4. Materials and Methods

### 4.1. Cell Culture and Differentiation 

Human neuroblastoma SH-SY5Y cells (wild-type) (CLS Cat# 300154/p822_SH-SY5Y, RRID:CVCL_0019) were obtained from ATCC, USA. SH-SY5Y neuroblastoma cell lines stably expressing α-synuclein or its A53T mutant were produced in our previous work [26]. Undifferentiated cells were maintained in Basic Growth Media (DMEM/F12 (1:1) medium (Thermo Fisher Scientific, Waltham, MA, USA) supplemented with 2 mM L-glutamine, 10% fetal bovine serum (FBS, HyClone, Logan, UT, USA),100 U/mL penicillin, and 100 μg/mL streptomycin at 37 °C in a humidified atmosphere with 5% CO_2_. The cells were passaged by trypsinization every 4 to 5 days or when they became 70% to 80% confluent; the number of passages did not exceed 10–15.

For differentiation to dopaminergic neuronal phenotype, a two-step protocol was used [28,30]. On day 0, cells were seeded in 200 µL of Basic Growth Media at a density of 1 × 10^4^ cells per well on 96-well round plates covered with 0.02 mg/mL PDL (#P6407, Sigma, St. Louis, MO, USA). On day 1, the medium was exchanged to the differentiation medium #1 (DMEM with 2.5% FBS, 2 mM L-glutamine, 1% fetal bovine serum, 100 U/mL penicillin, 100 μg/mL streptomycin, and 10 μM RA); cells were protected from light. On day 3, the medium was exchanged to the differentiation medium #2 (DMEM/F12 (1:1) with B-27, 2 mM L-glutamine, 100 U/mL penicillin, 100 μg/mL streptomycin, 50 ng/mL BDNF, and 10 μM RA); cells were protected from light. For the experiments, the cells were treated with 300 nM of each peptide on day 9. Cell death and cell viability assays were performed on day 10, and cells for real-time PCR were harvested on day 10.

### 4.2. Cell Counting Kit-8 (CCK-8) Cell Viability Assay

The cell viability was examined using Cell Counting Kit-8 (CCK-8) assay kits (#96992, Sigma, Hiroshima, Japan) according to the manufacturer’s instructions. CCK8 contains the tetrazolium salt WST-8, which is metabolized by living cells to form a soluble in cell culture medium dye formazan, which can be detected by optical density. The amount of this dye is directly proportional to the number of living cells. On day 10 of differentiation after 24 h treatment of cells with peptides, 10 µL of CCK-8 reagent was added to each well and incubated at 37 °C in a humidified atmosphere with 5% CO_2_ for 3 h. The absorbance at 450 nm was determined, using an Infinite F50 Tecan microplate photometer (Tecan, Grödig, Austria). Three wells of cells were used for each condition in every independent experiment. The cell viability of untreated cells was presented as a percentage of untreated control SH-SY5Y cells. The cell viability in peptide-treated cells was presented as a percentage of that in control or untreated cells in parallel experiments.

### 4.3. Propidium Iodide (PI) Staining Cell Death Assay

Cell death was examined using a PI staining assay (Invitrogen, Thermo Fisher, #P3566, Waltham, MA, USA). Cells plated and differentiated as described above in 96-well plates were 24 h pretreated with 300 nM Ms 9a-1 or 300 nM APHC3 peptide. PI was added into the culture medium with the final concentrations of 2 µg/mL. Cells were incubated at 37 °C for 10 min. For each condition in every independent experiment, three wells of cells were used. Absorbance fluorescence intensity was assessed using the NOVOstar microplate reader (BMG Labtech, GmbH, Ortenberg, Germany). Cell death was presented as a percentage of PI-positive cells to all cells identified by freeze-thawing in the same wells.

### 4.4. Immunofluorescence Imaging

Immunofluorescence was used to visualize differentiated cells. Cells grown on poly-d-lysine were fixed with 4% paraformaldehyde in PBS. After washing in PBS, cells were permeabilized in 0.1% Triton X-100/PBS/0.1% FBS for 30 min at 4 °C and again washed twice with PBS/0.1% FBS. Then samples were incubated with 0.1% Triton X-100, PBS, and 1% FBS in PBS for 2 h. To identify nuclei, cells were incubated with 1 µg/mL Hoechst 33342 (Thermo Fisher Scientific). To assess cell differentiation, cells were incubated with anti-neural cell adhesion molecule monoclonal antibodies 1:200 in PBS containing 1% FBS and Triton X-100 (NCAM; 56C04; Thermo Fisher Scientific). Then, samples were treated for 1 h with rabbit anti-mouse IgG H + L conjugated with Alexa Fluor^®^ 546 (1:1000; Thermo Fisher Scientific), washed PBS containing 1% FBS and Triton X-100, and then stained with antibodies to β-III tubulin conjugated with Alexa Fluor 647 (BioLegend; 1:100). The images were obtained using an Eclipse Ti-E microscope with an A1 confocal module (Nikon Corporation, Tokyo, Japan) and a CFI Plan Apo VC 20×/0.75 objective. Data were visualized using Nikon proprietary software (NIS-Elements).

### 4.5. RNA Extraction and cDNA Synthesis

Total RNA was extracted from frozen cell pellets with TRIzol™ Reagent (Life Technologies, Carlsbad, CA, USA) and the following chloroform phase separation and ethanol precipitation at −20 °C. Washed RNA was dissolved in Steril RNAse-free water (Thermo Fisher Scientific, Waltham, MA, USA). The final RNA concentration was measured using a spectrophotometer. To prevent RNA aggregation, total RNA samples were heated at 65 °C for 1–2 min before cDNA synthesis; samples were immediately used in cDNA synthesis. DNAse treatment was performed with RQ1 RNAse-free DNAse (Promega, Madison, WI, M6101). First-strand cDNA was synthesized from 1 µg total RNA using Mint Kit (Evrogen, Moscow, Russia) according to the manufacturer’s instructions.

### 4.6. Quantitative PCR

Real-time PCR (qPCR) was performed using SsoAdvanced Universal SYBR Green Supermix (Bio-Rad, Hercules, CA, USA) in a 7500 real-time PCR system (Applied Biosystems, Waltham, MA, USA). We used the following primers:

ASIC1a: fwd 5′-AGCGGCTGTCTCTGAAGC-3′, rev 5′-AGCTCCCCAGCATGATACAG-3′; TRPV1: fwd 5′-CGCTGATTGAAGACGGGAAG-3′, rev 5′-CAGGAGGATGTAGGTGAGAATTAC-3′; TRPA1: fwd 5′-GGAACACTGCACTTCACTTTG-3′, rev 5′-CATGCATTCAGGGAGGTATTC-3′; β-actin: fwd 5′-CCACGAAACTACCTTCAACTCC-3′, rev 5′-TCGTCATACTCCTGCTTGCTGATCC-3′; TH: fwd 5′-TCATCACCTGGTCACCAAGTT-3′, rev 5′-GGTCGCCGTGCCTGTACT-3′; SNCA: fwd 5′-GTGGTCTATTTCTCCCTTCAATC-3′, rev 5′-CATCTTCTACACTGCTTAGTTCC-3′; BAX: fwd 5′-TCAGGATGCGTCCACCAAGAAG-3′, rev 5′-TGTGTCCACGGCGGCAATCATC-3′; Bcl-2: fwd 5′-AGATGTCCAGCCAGCTGCACC-3′, rev 5′-GGCATGTTGACTTCACTTGTG-3′.

The relative mRNA levels were calculated according to the 2^−ΔΔCT^ method based on the threshold cycle (C_T_) values. Undifferentiated cells or cells without any treatment served as controls. The quantity of transcripts of the control sample was normalized by the housekeeping gene β-actin; the experimental samples were compared with it. Samples without cDNA served as negative controls for each gene. All reactions were performed in triplicate. The threshold cycle (C_T_) is the cycle at which the fluorescence level reaches a threshold value, these data were obtained using the 7500 Real-Time PCR System. ΔCT method: ΔC_T_ = (C_T_ gene of interest − C_T_ β-actin); 2^−ΔCT^ is the measure of the mRNA expression level in the sample normalized to the housekeeping gene β-actin. ΔΔC_T_ method: ΔΔC_T_ = ((C_T_ gene of interest − C_T_ β-actin) sample cDNA) − ((C_T_ gene of interest—C_T_ β-actin) control cDNA); 2^−ΔΔCT^ is the relative expression value that is the measure of the mRNA expression in the sample normalized to the control sample. The real-time PCR protocol was as follows: denaturation (95 °C, 10 min), 40 cycles of denaturation (95 °C, 15 s), annealing (60 °C, 1 min), and elongation (72 °C, 30 s). The resulting PCR products created single bands of appropriate sizes in agarose gel electrophoresis.

### 4.7. Data Presentation and Statistical Analysis 

The statistical analysis was performed with GraphPad Prism 8.0.1. Measurement data are expressed as mean ± SD. *p* < 0.05 was considered to indicate a statistically significant difference. The significance of the data differences was determined with a one-way analysis of variance (ANOVA) followed by Dunnett’s multiple comparisons test.

## 5. Conclusions

We developed a new neuronal-like cell model system based on differentiated to dopaminergic neuron-like SH-SY5Y cells. We used three different types of cells: native neuroblastoma SH-SY5Y, cells overexpressing alpha-synuclein, and its mutant A53T, which is prone to aggregate. Differentiated cells significantly increased the susceptibility to apoptosis, but overexpression of alpha-synuclein, but not its A53T mutant, significantly reduced cell death. Expression of alpha-synuclein A53T mutant leads to a marked increase in the expression of the pro-apoptotic protein BAX, which could be attributed to the ability of mutant synuclein to form cytotoxic aggregates. Therefore, the expression of alpha-synuclein is a native neuroprotective mechanism, but in dopaminergic-like neurons, it causes a reduction in tyrosine hydroxylase expression, which may lead to a drop in dopamine production. Taken together, overexpression of alpha-synuclein may provide a model for the early stages of PD when alpha-synuclein allows neurons to survive but reduces dopamine production. Expression of mutant alpha-synuclein A53T mimics a genetic variant of PD when cytotoxic aggregates reduce both neuronal survival and dopamine production. We found expression of ASIC1a, TRPV1, and TRPA1 channels in dopaminergic neuron-like cells derived from the neuroblastoma SH-SY5Y. Peptide modulators of TRPV1 and TRPA1 significantly decreased the susceptibility of all cell lines to apoptosis. Therefore, modulators of TRPA1 and TRPV1 have the potential for the development of new therapeutic agents for the treatment of neurodegenerative diseases. In addition, the cellular model proposed and described in our work can be used for further studies of the involvement of ASIC1a, TRPV1, and TRPA1 channels in the neurodegeneration process, as well as for testing new ligands of these channels.

## Figures and Tables

**Figure 1 ijms-25-00368-f001:**
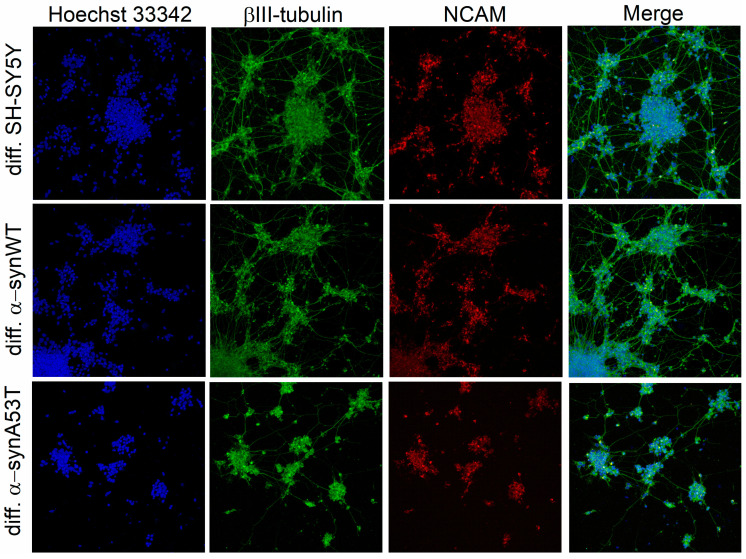
Immunofluorescence shows detectable levels of β-III-tubulin and NCAM in differentiated cells of all cell lines. Individual channel intensities were adjusted for appropriate visualization. The nucleus was stained with Hoechst 33342 (blue); the differentiated cells were stained with antibodies to βIII-tubulin (green) and anti-neural cell adhesion molecule NCAM (red). Confocal fluorescent microscope images taken at 20× magnification.

**Figure 2 ijms-25-00368-f002:**
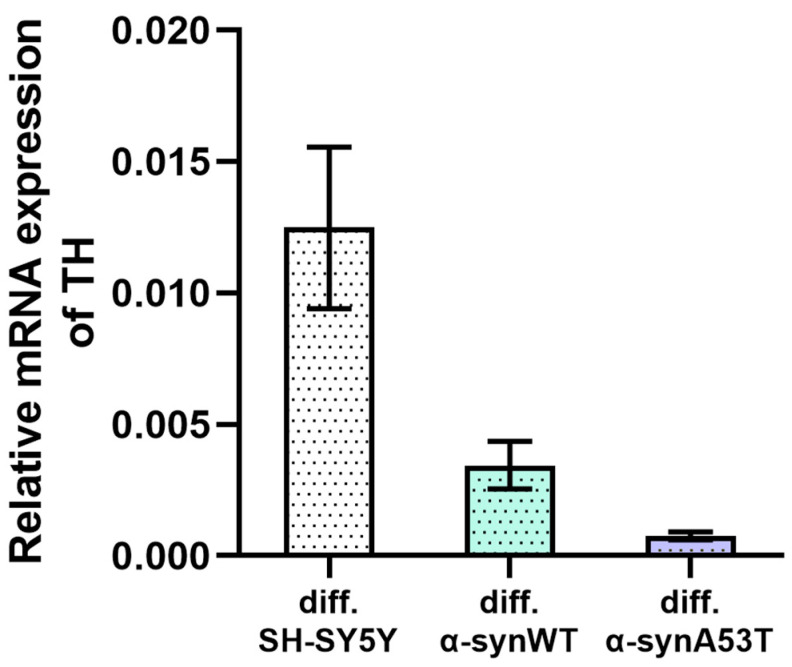
The relative levels of TH mRNA transcripts at d10 (differentiated cells) were analyzed by RT-qPCR and normalized to the housekeeping gene β-actin (ΔC_T_ method). Expression of TH mRNA in undifferentiated cells of the corresponding lines was not detected. Data are shown as mean ± SD.

**Figure 3 ijms-25-00368-f003:**
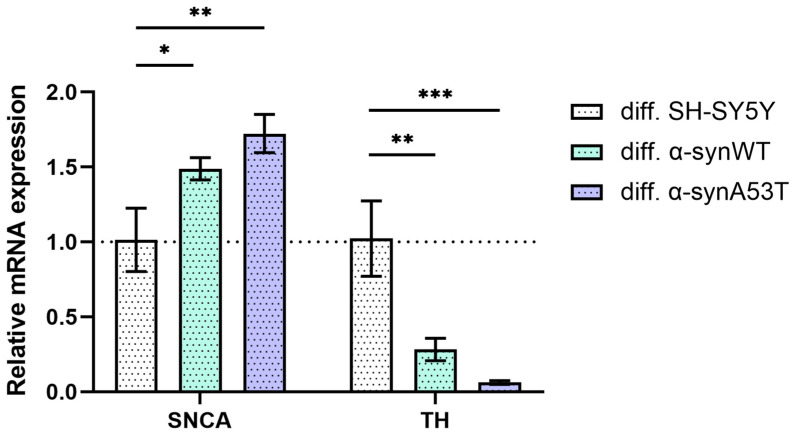
The relative levels of SNCA and TH mRNA transcripts at d10 (differentiated cells) in control (SH-SY5Y cells), α-synWT, and α-synA53T were analyzed by RT-qPCR and normalized to the housekeeping gene β-actin and the corresponding mRNA of SH-SY5Y cells (ΔΔC_T_ method). Data are shown as mean ± SD (data are from 3 independent experiments, with 3 technical replications each). Statistical analysis was performed using the one-way ANOVA test, followed by Dunnett’s multiple comparisons test; *—*p* < 0.05, **—*p* < 0.01, ***—*p* < 0.001.

**Figure 4 ijms-25-00368-f004:**
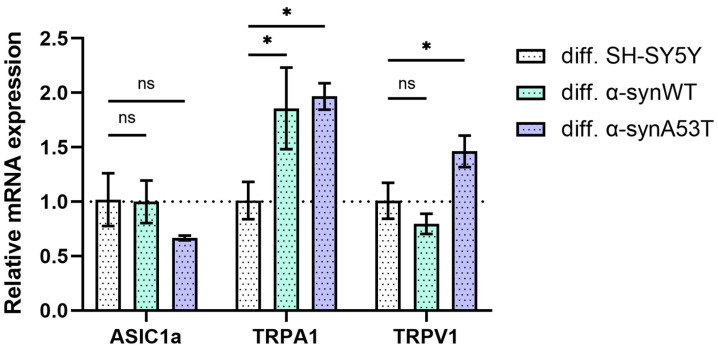
The relative levels of ASIC1a, TRPA1, and TRPV1 mRNA transcripts at d10 (differentiated cells in control (SH-SY5Y cells), α-synWT, and α-synA53T were analyzed by RT-qPCR and normalized to the housekeeping gene β-actin and the corresponding mRNA of differentiated SH-SY5Y cells (ΔΔC_T_ method). Data are shown as mean ± SD (data are from 3 independent experiments, with 3 technical replications each). Statistical analysis was performed using the one-way ANOVA test followed by Dunnett’s multiple comparisons test; *—*p* < 0.05. ns—not significant.

**Figure 5 ijms-25-00368-f005:**
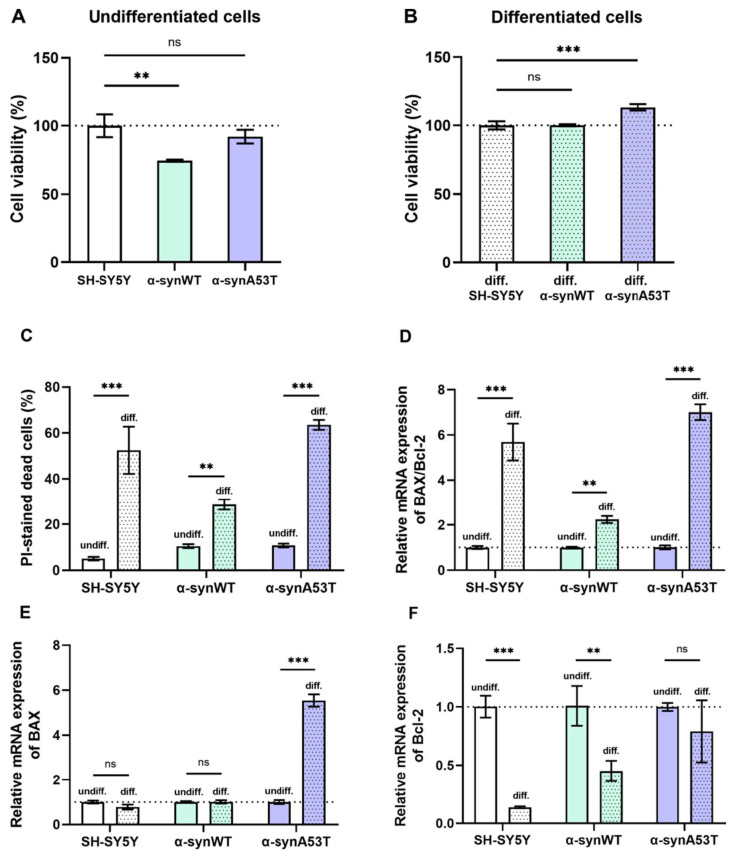
(**A**,**B**) Changes in cell viability during differentiation normalized to the cell viability of corresponding control SH-SY5Y cells. (**C**) The cell death in undifferentiated and differentiated cells. (**D**–**F**) Relative mRNA expression levels were analyzed by RT-qPCR and normalized to the housekeeping gene β-actin and the corresponding mRNA of undifferentiated cells (ΔΔC_T_ method). (**D**) The BAX/Bcl2 expression ratio in the undifferentiated and differentiated SH-SY5Y (white color), α-synWT (green color), and α-synA53T cells (violet color). The BAX/Bcl2 ratio was significantly increased in differentiated cells compared to undifferentiated cells of the corresponding cell line. Relative mRNA expression levels of BAX (**E**) and Bcl-2 (**F**) detected in the undifferentiated and differentiated SH-SY5Y, α-synWT, and α-synA53T cells. Data are shown as mean ± SD (data are from 3 independent experiments, with 3 technical replications each). Statistical analysis was performed using the one-way ANOVA test followed by Dunnett’s multiple comparisons test; **—*p* < 0.01, ***—*p* < 0.001. ns—not significant.

**Figure 6 ijms-25-00368-f006:**
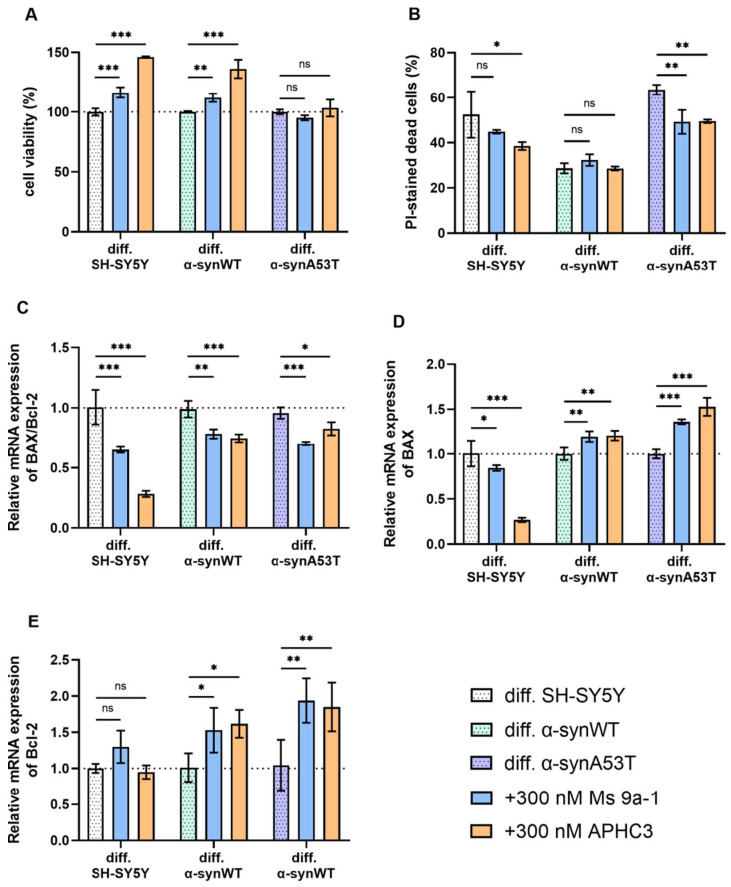
(**A**) The cell viability of differentiated cells after 24 h peptide treatment (300 nM for each peptide) normalized to the cell viability of corresponding untreated cells. (**B**) The cell death in differentiated cells after 24 h peptide treatment (300 nM each peptide). Cell death is represented as the percentage of PI-positive cells to all cells detected by freeze-thawing in the same samples. (**C**–**E**) Relative levels of mRNA expression were analyzed by RT-qPCR and normalized to the housekeeping gene β-actin and the corresponding mRNA of untreated differentiated cells (ΔΔC_T_ method). (**C**) The BAX/Bcl2 expression ratio in differentiated SH-SY5Y, α-synWT, and α-synA53T cells after 24 h peptide treatment (300 nM each peptide). The BAX/Bcl2 ratio was significantly decreased in peptide-treated cells compared to untreated cells. (**D**) Bax and (**E**) Bcl-2 in the differentiated SH-SY5Y, α-synWT, and α-synA53T cells after 24 h peptide treatment (300 nM each peptide). Data are shown as mean ± SD (data are from 3 independent experiments, with 3 technical replications each). Statistical analysis was performed using the one-way ANOVA test followed by Dunnett’s multiple comparisons test; *—*p* < 0.05, **—*p* < 0.01, ***—*p* < 0.001. ns—not significant.

**Figure 7 ijms-25-00368-f007:**
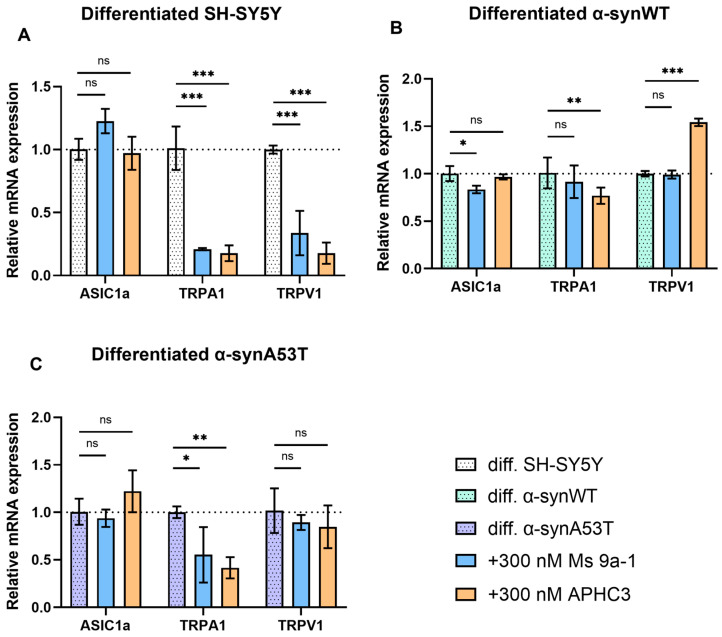
Ms 9a-1 and APHC3 peptides modify the expression levels of ASIC1a, TRPA1, and TRPV1 channels. The relative levels of ASIC1a, TRPA1, and TRPV1 mRNA transcripts at d10 in differentiated cells treated with 300 nM Ms 9a-1 or 300 nM APHC3 for 24 h were detected and analyzed by RT-qPCR and normalized to the housekeeping gene β-actin and the corresponding mRNA of untreated differentiated cells (ΔΔC_T_ method). (**A**) Differentiated SH-SY5Y cells. (**B**) Differentiated α-synWT cells. (**C**) Differentiated α-synA53T cells. Data are presented as mean ± SD (data are from 3 independent experiments, with 3 technical replications each). Statistical analysis was performed using the one-way ANOVA test followed by Dunnett’s multiple comparisons test; *—*p* < 0.05, **—*p* < 0.01, ***—*p* < 0.001 versus the control group. ns—not significant.

**Table 1 ijms-25-00368-t001:** Difference in differentiated cell lines α-synWT and α-synA53T compared to native SH-SY5Y. The direction of the arrows indicates an increase or decrease in gene expression levels or cell viability.

	Diff. α-synWT	Diff. α-synA53T
cell viability	ns	↑ 13% ***
SNCA1 ^1^	↑ 47% *	↑ 70% **
TH	↓ 72% **	↓ 94% ***
ASIC1a	ns	ns
TRPA1	↑ 84% *	↑ 95% *
TRPV1	ns	↑ 45% *

^1^ Relative level of corresponding gene expression. *—*p* < 0.05, **—*p* < 0.01, ***—*p* < 0.001 versus control group, ns—not significant.

**Table 2 ijms-25-00368-t002:** Changes in cell lines induced by Ms9a-1 and APHC3 treatment. The direction of the arrows indicates an increase or decrease in gene expression levels or other parameters.

	Diff. SH-SY5Y + Ms 9a-1	Diff. SH-SY5Y + APHC3	Diff. α-synWT + Ms 9a-1	Diff. α-synWT + APHC3	Diff. α-synA53T + Ms 9a-1	Diff. α-synA53T + APHC3
cell viability	↑ 16.37% ***	↑ 46.05% ***	↑ 11.97% **	↑ 36.12% ***	ns	ns
cell death	ns	↓ 13.97% *	ns	ns	↓ 14.35 **	↓ 14.06 **
BAX/Bcl-2	↓ 35.4% ***	↓ 72% ***	↓ 21% **	↓ 24.5% ***	↓ 26.6% ***	↓ 13.7% *
ASIC1a	ns	ns	↓ 16% *	ns	ns	ns
TRPA1	↓ 79% ***	↓ 82.5% ***	ns	↓ 24% **	↓ 44.7% *	↓ 58% **
TRPV1	↓ 57% ***	↓ 77% ***	ns	↑ 54% ***	ns	ns

*—*p* < 0.05, **—*p* < 0.01, ***—*p* < 0.001 versus control group, ns—not significant.

## Data Availability

All data are contained within the article.

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
