# Peer review of "Modulation of TRPV1 and TRPA1 Channels Function by Sea Anemones’ Peptides Enhances the Viability of SH-SY5Y Cell Model of Parkinson’s Disease"

_ijms, 2023, doi:10.3390/ijms25010368_

Round 1

Reviewer 1 Report

Comments and Suggestions for Authors

In the manuscript entitled “Modulation of TRPV1 and TRPA1 channels function by sea anemones’ peptides enhances the viability of SH-SY5Y cell model of Parkinson's disease’’ Kolesova and her colleagues, investigated the modulatory properties of two peptides, Ms9a-1 and APHC3,   on the function of TRPA1 and TRPV1 channels.

The manuscript is interesting, however important modifications would need to be made to be considered valid and it cannot be accepted in this form.

The manuscript is not uniform, it is as if there were parts of different texts, not homogeneous with each other. The text must be rewritten, and major editing work should be done to unify the text. An element of confusion in the text is that not all the molecules/substances mentioned in the work have full names; the full name of all the substances mentioned in the work should be indicated the first time it is mentioned, and the acronym in brackets.

The introduction should be clearer and more homogeneous, and some concepts should be extended, for example, the implication of TRPV receptors and ASICs in Parkinson's disease should be better explained, a comment on the nature of the peptides examined, the importance of the possible effect in humans and experimental models. Parts of the text currently under discussion could be moved to the introduction.

The results are very confusing, they should be rewritten and explained better and with greater accuracy to be truly convincing:

-the order of description of the results for the different molecules examined is different from the order of the same in the figures, this creates a lot of confusion in reading.

-each figure has a different layout from the others (position of titles, fonts, colors, dimensions and acronyms used), and some are also deformed; the layout of the figures should be the same for all, the titles in the x and y axis should be uniform,

- sometimes the results of the text do not agree with the graphs of the figures or refer to figures that are not present in the manuscript (such as figure 2B and figure 3A and 3B, 4A and 4B), therefore it is not possible to judge.

- in the result of Figure 1 the caption should be modified, indicating all the molecules present in the photos, the magnification used, etc.

-in the result of Figure 2 it is not clear whether a real-time PCR or a reverse transcriptase PCR was performed, and in general, in the text there is confusion in the use of the acronyms referring to the two methods.

-the result for TH in Figure 2 is the same as shown in Figure 3, why is it described twice?

- points 2.4, 2.5 and 2.6 are unclear

the discussion is not aligned with the results obtained; many sentences should be moved to the introduction paragraph.

The methods should be more exhaustive, the usual problem with acronyms, (for example incubation times of primary antibodies used in immunofluorescence, Tween or Triton? It seems strange that they use both); they should describe ΔCT and ΔΔCT methods, often indicated in the results.

Author Response

We are grateful to the reviewer for detailed analysis of our work and apologize for any discrepancies in the text. We have edited the text considerably, removing inconsistencies and trying to describe our work in a more acceptable format. Furthermore, we have tried to answer all questions in detail and have included a graphical summary of the work done to help understand the overall conclusion of the manuscript. Below we provide answers to all the questions raised.  

Reviewer 1

In the manuscript entitled “Modulation of TRPV1 and TRPA1 channels function by sea anemones’ peptides enhances the viability of SH-SY5Y cell model of Parkinson's disease’’ Kolesova and her colleagues, investigated the modulatory properties of two peptides, Ms9a-1 and APHC3,   on the function of TRPA1 and TRPV1 channels.

The manuscript is interesting, however important modifications would need to be made to be considered valid and it cannot be accepted in this form.

The manuscript is not uniform, it is as if there were parts of different texts, not homogeneous with each other. The text must be rewritten, and major editing work should be done to unify the text. An element of confusion in the text is that not all the molecules/substances mentioned in the work have full names; the full name of all the substances mentioned in the work should be indicated the first time it is mentioned, and the acronym in brackets.

The introduction should be clearer and more homogeneous, and some concepts should be extended, for example, the implication of TRPV receptors and ASICs in Parkinson's disease should be better explained, a comment on the nature of the peptides examined, the importance of the possible effect in humans and experimental models. Parts of the text currently under discussion could be moved to the introduction.

The results are very confusing, they should be rewritten and explained better and with greater accuracy to be truly convincing:

-the order of description of the results for the different molecules examined is different from the order of the same in the figures, this creates a lot of confusion in reading.

We have substantially rewritten the Results section to make it as clear as possible. In addition, we have added the study design to the graphical abstract, and we have also summarized the main results in tabular form to visualize the changes observed in the cell lines used in the work.

-each figure has a different layout from the others (position of titles, fonts, colors, dimensions and acronyms used), and some are also deformed; the layout of the figures should be the same for all, the titles in the x and y axis should be uniform,

We are grateful for the comments, the design of the figures has been brought to a more consistent style, including style, colors, and captions have been unified.

- sometimes the results of the text do not agree with the graphs of the figures or refer to figures that are not present in the manuscript (such as figure 2B and figure 3A and 3B, 4A and 4B), therefore it is not possible to judge.

We apologize for these annoying inaccuracies, we have corrected the numbering of the figures to match the text, and we have corrected typos in the text.

118 (Figure 3A and 2B) has been changed to (Figure 3)

123 added at the end of the sentence (Figure 3)

144 (Figure 4A, B) replaced with (Figure 4)

145 (Figure 2A) replaced with (Figure 4)

146 (Figure 4B) replaced with (Figure 4)

148 (Figure 4A, B) replaced with (Figure 4)

162 replaced in brackets (neuroblastoma SH-SY5Y) to (undifferentiated) to match the captions in the figures and to match all lines, not just the native line; and in (dopaminergic neuron-like cells) to (dopaminergic neuron-like cells, differentiated)

164 (Figure 3A) changed to (Figure 5A)

223 - changed α-synA53T to α-synWT

224 - changed α-synWT to α-synA53T (mixed up, viability does not change in mutants)

240 - described percentages (by 35.4% with Ms 9a-1 and 72% with APHC3)

242 (Figure 6E) changed to (Figure 6D, E)

- in the result of Figure 1 the caption should be modified, indicating all the molecules present in the photos, the magnification used, etc.

Thank you for your comment, the figure caption has been corrected to include all molecules used.

95 Figure 1. Immunofluorescence shows detectable levels of β-III-tubulin and NCAM in differentiated cells of all cell lines. Individual channel intensities were adjusted for appropriate visualization. The nucleus was stained with Hoechst 33342 (blue), the differentiated cells were stained with antibodies to βIII-tubulin (green) and anti-neural cell adhesion molecule NCAM (red). Confocal fluorescent microscope images taken at 20x magnification.

-in the result of Figure 2 it is not clear whether a real-time PCR or a reverse transcriptase PCR was performed, and in general, in the text there is confusion in the use of the acronyms referring to the two methods.

Thank you for your comment, total RNA was first transcribed into complementary DNA (cDNA). Then was performed real-time PCR (qPCR) from cDNA templates in all cases, we have corrected the caption of the figure.

-the result for TH in Figure 2 is the same as shown in Figure 3, why is it described twice?

In Figure 2, we present "absolute" TH expression values in all cell lines (since undifferentiated cells did not show TH expression, we normalized the expression values to the housekeeping gene β-actin (ΔCT method)) to show that the phenotype of the cells changes during differentiation. In Figure 3, we have compared the expression in transfected synuclein lines (α-synWT and α-synA53T) with native (SH-SY5Y cells), normalized to the housekeeping gene β-actin and the TH mRNA of SH-SY5Y cells (ΔΔCT method) to show that the transfected lines have differences in phenotype. If you think that these data should not be shown, we will, of course, remove Figure 2.

- points 2.4, 2.5 and 2.6 are unclear

the discussion is not aligned with the results obtained; many sentences should be moved to the introduction paragraph.

We have revised the text to clarify it.

The methods should be more exhaustive, the usual problem with acronyms, (for example incubation times of primary antibodies used in immunofluorescence, Tween or Triton? It seems strange that they use both); they should describe ΔCT and ΔΔCT methods, often indicated in the results.

We regret any inaccuracies that may have occurred. We have tested different detergents in the preliminary stages, but now the method description is clear and more detailed.

We have added the description of the ΔCT and ΔΔCT to the Methods.

The threshold cycle (CT) is the cycle at which the fluorescence level reaches a threshold value, these data were obtained using The 7500 Real-Time PCR System. ΔCT method: ΔCT = (CT gene of interest - CT β-actin). 2 ΔCT is the measure of the mRNA expression level in the sample normalized to the housekeeping gene β-actin. ΔΔCT method: ΔΔCT = [(CT gene of interest - CT β-actin) sample cDNA] — [(CT gene of interest - CT β-actin) control cDNA]; 2 ΔΔCT = Relative expression value that is the measure of the mRNA expression in the sample normalized to the control sample.  

Thus for Figure 3, 4, 5D-F,  ΔΔCT method: ΔΔCT = [(CT gene of interest- CT β-actin) cDNA of differentiated cells]—[(Ct gene of interest-Ct internal control) cDNA of undifferentiated SH-SY5Y cells]; 2 −ΔΔCT = Relative expression.

For Figure 6C-E, 7 ΔΔCT method: ΔΔCT = [(CT gene of interest- CT β-actin) cDNA of peptide-treated differentiated cells]—[(Ct gene of interest-Ct internal control) cDNA of untreated differentiated cells of the corresponding cell line]; 2 −ΔΔCT = Relative expression

Reviewer 2 Report

Comments and Suggestions for Authors

After thoroughly reviewing the manuscript titled "Modulation of TRPV1 and TRPA1 channels function by sea anemones’ peptides enhances the viability of SH-SY5Y cell model of Parkinson's disease" here are my detailed suggestions for revision:

  1. Experimental Design and Methodology:
    • The cell culture and differentiation methods are well-described. However, consider including more detailed information about the conditions under which the cells were maintained, such as temperature, CO2 concentration, and humidity, as these factors could impact the results.
  2. Data Analysis and Presentation:
    • The manuscript could benefit from a more in-depth explanation of the statistical methods used, particularly the rationale behind choosing specific statistical tests.
    • For the gene expression data, it would be beneficial to include additional statistical analyses that could account for multiple testing corrections, given the number of comparisons being made.
  3. Results Interpretation:
    • Provide a more detailed discussion on the implications of the findings related to the modulation of TRPV1 and TRPA1 channels in Parkinson's disease.
    • Discuss how the findings relate to the current understanding of Parkinson's disease pathophysiology and how they might contribute to developing new therapeutic strategies.
  4. Figures and Supplementary Material:
    • Ensure that all figures are clearly labeled and have high-resolution images. Consider including additional graphical representations such as bar graphs or scatter plots to better illustrate the key findings.
    • Supplementary material, if any, should be directly referenced in the manuscript and easily accessible for readers.
  5. References and Citations:
    • Verify that all references are current and accurately reflect the latest research in the field. Include recent studies to demonstrate the manuscript's relevance to current research.

o   Consider adding more recent references to support statements, especially in rapidly evolving areas of the field. Where possible, include recent studies to demonstrate the manuscript's alignment with current research trends. In particular, consider including additional references to support the discussion and to provide context to the study’s findings. I suggest adding data related to recent bulk transcriptomics studies which could represent a strong substrate to enforce the role of described molecular mechanisms, such as the recent PMID: 36490268 and PMID: 27737651.

  1. Technical Limitations and Biases:
    • Discuss the potential limitations of using SH-SY5Y cells as a model for Parkinson's disease, including how well this cell line represents the complexity of the disease in humans.
    • Address potential biases in the experimental design and how they were mitigated.
  2. Conclusion and Future Work:
    • Strengthen the conclusion by summarizing key findings more effectively, emphasizing their implications for Parkinson's disease research and potential therapeutic applications.
    • Suggest future research directions, including potential in vivo studies or investigations into other neurodegenerative diseases.
  3. Ethical Considerations:
    • If not already included, add a brief section discussing the ethical considerations of using cell lines in research, particularly concerning donor consent and the use of genetically modified organisms.
Comments on the Quality of English Language

The English should be improved.

Author Response

We are grateful to the reviewer for his attention and detailed analysis of our work. We have tried to answer all questions in detail and have included a graphical summary of the work done to help understand the overall conclusion of the manuscript. Below we provide answers to all the questions raised.

Reviewer 2

After thoroughly reviewing the manuscript titled "Modulation of TRPV1 and TRPA1 channels function by sea anemones’ peptides enhances the viability of SH-SY5Y cell model of Parkinson's disease" here are my detailed suggestions for revision:

  1. Experimental Design and Methodology:
    • The cell culture and differentiation methods are well-described. However, consider including more detailed information about the conditions under which the cells were maintained, such as temperature, CO2 concentration, and humidity, as these factors could impact the results.
    •  

Details about the conditions under which the cells were maintained are provided in Material and Methods section.

  1. Data Analysis and Presentation:
    • The manuscript could benefit from a more in-depth explanation of the statistical methods used, particularly the rationale behind choosing specific statistical tests.
    • For the gene expression data, it would be beneficial to include additional statistical analyses that could account for multiple testing corrections, given the number of comparisons being made.
    •  

We performed an additional Dunnett's test (post One-Way ANOVA) including correction for multiple comparisons, this is indicated in the methods and in each figure caption where required.

One-Way ANOVA test is used to determine if there are statistically significant differences between two or more groups on one independent variable. Our dependent variable is expression and the independent variable is peptide. For example, we compare the expression in untreated cells of one cell line with the expression of Ms 9a-1 peptide treated cells of the same cell line. In this way we find out if different genes respond differently in that particular line. We repeat this way for each line. Then we repeat this way for the second APHC3 peptide. The Dunnett's test that follows takes into account the possibility of scatter in the groups being compared, it compares each of our experimental averages to the control average. This gives a more accurate scatter value, more power to detect differences. This is a particularly useful method to analyse studies having control groups, based on modified t-test statistics (Dunnett's t-distribution). It is a powerful statistic tool and, therefore, can discover relatively small but significant differences among groups or combinations of groups. The Dunnett test is used by researchers interested in testing two or more experimental groups against a single control group.

  1. Results Interpretation:
    • Provide a more detailed discussion on the implications of the findings related to the modulation of TRPV1 and TRPA1 channels in Parkinson's disease.
    • Discuss how the findings relate to the current understanding of Parkinson's disease pathophysiology and how they might contribute to developing new therapeutic strategies.
    •  

We have modified the discussion according to the comments. Some concepts have been extended, for example, the implication of TRPV receptors and ASICs in Parkinson's disease and possible links between our findings and current understanding of the pathophysiology of Parkinson's disease.

  1. Figures and Supplementary Material:
    • Ensure that all figures are clearly labeled and have high-resolution images. Consider including additional graphical representations such as bar graphs or scatter plots to better illustrate the key findings.
    • Supplementary material, if any, should be directly referenced in the manuscript and easily accessible for readers.
    •  

We are particularly grateful to the reviewer for this comment, as the added graphical representation will enrich the manuscript for readers. We have also summarised the main results in tabular form to visualize the changes observed in the cell lines used in the paper.

  1. References and Citations:
    • Verify that all references are current and accurately reflect the latest research in the field. Include recent studies to demonstrate the manuscript's relevance to current research.
    • Consider adding more recent references to support statements, especially in rapidly evolving areas of the field. Where possible, include recent studies to demonstrate the manuscript's alignment with current research trends. In particular, consider including additional references to support the discussion and to provide context to the study’s findings. I suggest adding data related to recent bulk transcriptomics studies which could represent a strong substrate to enforce the role of described molecular mechanisms, such as the recent PMID: 36490268 and PMID: 27737651.

We have included references to modern directions in the field, demonstrating the relevance of the manuscript to current research.

Bulk transcriptomic approaches, including single-cell data, are also appropriate to identify differences in gene expression patterns and the functional biological processes to which they are linked in PD pathogenesis (38020762). One possibility for how the expression of genes involved in the pathogenesis of PD may be altered is at the level of transcriptional regulation, as has been shown for cerebral cavernous malformation disease, which also exists in sporadic and genetic forms (27737651).

  1. Technical Limitations and Biases:
  • Discuss the potential limitations of using SH-SY5Y cells as a model for Parkinson's disease, including how well this cell line represents the complexity of the disease in humans.
  • Address potential biases in the experimental design and how they were mitigated.

Our study has a complex initial design to obtain a suitable cellular model of Parkinson's disease, which requires cautious interpretation of the results. The use of SH-SY5Y cells as a model for Parkinson's disease has a number of potential limitations, such as significant dependence of cell survival on culture conditions, lack of natural neuronal networks and interactions with glial cells, etc. Therefore, several endpoints (morphological assessment of differentiation, viability assays, metabolic assays) were chosen to obtain clear and unambiguous results, but further validation of our main findings at different levels and models is needed.

  1. Conclusion and Future Work:
  • Strengthen the conclusion by summarizing key findings more effectively, emphasizing their implications for Parkinson's disease research and potential therapeutic applications.
  • Suggest future research directions, including potential in vivo studies or investigations into other neurodegenerative diseases.
  •  

We are grateful for the comment and have described future research directions and potential links of our research to the development of therapeutic approaches in the Conclusions section.

  1. Ethical Considerations:
  • If not already included, add a brief section discussing the ethical considerations of using cell lines in research, particularly concerning donor consent and the use of genetically modified organisms.
  •  

An ethics committee opinion is not required for this study; we have clarified this in the Institutional Review Board Statement section.

Round 2

Reviewer 1 Report

Comments and Suggestions for Authors

The authors followed suggestions and comments, so I think that the current manuscript is suitable for publication in International Journal Of Molecular Sciences.

I don’t have other specific queries.

Reviewer 2 Report

Comments and Suggestions for Authors

The authors addressed all suggested points.

Comments on the Quality of English Language

The English was improved.